# Pine-Oil-Derived Sodium Resinate Inhibits Growth and Acid Production of *Streptococcus mutans* In Vitro

**DOI:** 10.3390/dj12020040

**Published:** 2024-02-17

**Authors:** Otto Rajala, Matias Mäntynen, Vuokko Loimaranta

**Affiliations:** Institute of Dentistry, University of Turku, Lemminkäisenkatu 2, 20520 Turku, Finland

**Keywords:** *Streptococcus mutans*, sodium resinate, *Pinus sylvestris*, acid production, oral biofilm, caries, optical pH sensors

## Abstract

*S. mutans* is a key pathogen in dental caries initiation and progression. It promotes oral biofilm dysbiosis and biofilm acidification. Sodium resinate is a salt of pine-oil-derived resin which has antimicrobial properties. Pine-oil-derived resin consists of terpenes, diterpenes, and abietic acids. The aim of this study was to determine the effects of pine (*Pinus sylvestris*) oil resinate (RS) on growth and acid production of cariogenic *S. mutans* strains in planktonic form and biofilm. The *S. mutans* type strain NCTC10449 and clinical isolate CI2366 were grown on 96-well plates for testing of RS effects on growth and biofilm formation, and on plates with integrated pH-sensitive optical ensors for real-time measurements of the effects of RS on bacterial acid production. We found that even short-time exposure to RS inhibits the growth and acid production of *S. mutans* in the planktonic phase and biofilms. In addition, RS was able to penetrate the biofilm matrix and reduce acid production inside *S. mutans* biofilm. RS thus shows potential as a novel antibacterial agent against cariogenic bacteria in biofilm.

## 1. Introduction

Dental caries is the most common oral infectious disease. It is a polymicrobial disease where acidic bacterial metabolites cause demineralization of the tooth hard tissue [1,2]. The disease results from a dysbiosis of the oral microbiome due to an ecological imbalance caused by a combination of endogenous bacteria, dietary habits, oral hygiene, host genetics, and the amount and quality of saliva [1,3,4]. The cariogenic microbiome is rich in acid-producing bacteria [5], and among the most efficient acid producers are *Streptococcus mutans* bacteria [6,7]. *S. mutans* produces a higher amount of acids at a low pH compared with other Streptococci [8], and acidification of a biofilm is very rapid within *S. mutans* microcolonies [9].

On the tooth surfaces, microbes live in a biofilm, dental plaque, that protects them from saliva and other host defense mechanisms, increases their chances of attaching to tooth surfaces and increase their metabolic effectiveness [3,10]. Bacteria in biofilms produce a supporting matrix called the extracellular matrix (ECM) that promotes attachment and gives the biofilm structure and protection [11]. The matrix is mainly composed of proteins, extracellular DNA, and polysaccharides secreted by bacteria in the biofilm. *S. mutans* is efficient in producing extracellular polysaccharides that results in thick and sticky ECM in bacterial biofilms [7]. ECM can prevent antibiotics and cationic antimicrobials such as chlorhexidine from reaching the inner parts of biofilm, making them ineffective against bacterial action in mature biofilms [11]. ECM also prevents diffusion of bacteria-produced acids from the biofilm while the environmental components, such as salivary buffers, do not reach the inner parts of the biofilm structure. Thus, the microenvironment inside the biofilm can remain highly acidic despite the continuous flow of saliva and its high buffering capacity [7].

Several antimicrobial agents such as chlorhexidine, alcohol, iodine, fluoride, and xylitol are used in mouthwashes, chewing gum, and dentifrices, e.g., Refs. [12,13,14]. Mouthwashes have an important role in maintaining oral health in patients with challenges in tooth brushing and improving the effectiveness of tooth brushing in severe caries or periodontitis [12,13]. Chlorhexidine (CHX) has been the gold standard in clinically effective mouthwashes, but it has several possible side effects such as a burning sensation in the mouth, mucosal irritation, and an altering of taste [12,15,16]. In addition, bacteria can become resistant to CHX, and resistance to CHX has been linked to general antibiotic resistance as well [16]. This resistance highlights the need to develop new antimicrobial agents.

Coniferous plants consist of a number of genera and species found worldwide. They have been used as medicinal sources in many cultures for centuries. Indeed, compounds with antimicrobial, antioxidant, anti-inflammatory, and anticancerous properties have been described in various extracts of coniferous plants [17]. The coupling of conifer bark extract with silver nanoparticles and rosin-grafted cellulose nanocrystals are interesting examples of enhancing the antimicrobial effects of these phytochemicals [18,19]. One evident source of bioactive conifer compounds is oleoresin (pitch, resin), which conifers produce for defense against insects and physical damage [20]. Oleoresins consist of terpenoids with roughly equal parts turpentine and rosin, while rosin consists of non-water-soluble diterpene resin acids such as abietic acid, neoabietic acid, and dehydroabietic acid [20]. Conifers have long been recognized for their antimicrobial properties, and, for example, resin salves have been used in folk medicine for centuries in the treatment of burn injuries and skin wounds [21]. Coniferous rosin, rosin derivatives, or purified abietic acids are bacteriostatic and they are effective against gram-positive and gram-negative bacteria [21,22,23,24,25]. The exact phytochemical composition varies according to species and growth conditions, but at least pure abietic acid and dehydroabietic acid from pine rosin are also antimicrobial against oral gram-positive bacteria, such as *S. mutans* [26,27,28]. In addition, rosin derivatives are potential antiviral substances, and extracts of *Pinus sylvestris* and *Picea abies* oil inactivate human enveloped viruses, such as influenza A virus (IAV), respiratory syncytial virus, and severe acute respiratory syndrome coronavirus 2 (SARS-CoV-2) [29,30].

Overall rosin-derived molecules show great promise as antimicrobial agents, but their poor solubility in aqueous media restricts their wider use in dental products like mouthwashes. This study aimed to determine the effects of pine resinate (RS), a more water-soluble sodium salt of resin acid, on the growth and acid production of cariogenic *S. mutans* strains in developing and mature biofilm. The hypothesis was that RS inhibits *S. mutans* in planktonic form and also in biofilms.

## 2. Materials and Methods

Two *S. mutans* strains, type strain NCTC10449 and clinical isolate CI2366, were used in this study. The clinical isolate was isolated in an earlier study from the saliva of a non-xylitol-consuming female [31]. The identification is described in detail in the earlier study [31]. Briefly, bacteria were grown on Mitis Salivarius Bacitracin (MSB, Difco, Detroit, MI, USA) plates and identified based on colony morphology. The colonies were transferred to the blood and Mitis Salivarius (MS, Difco) plates to produce pure clinical strains. To verify the purity, the strains were grown in Brain Heart Infusion broth (BHI, Difco), followed by plate culturing on MS and blood agar (Orion Diagnostica, Espoo, Finland).

The bacteria were stored in frozen aliquots, and before each experiment fresh bacteria were grown in a liquid medium. The purity of the bacterial stock was confirmed before aliquoting and frozen by plating the bacteria on BHI agar plates. The resinate powder was resin acid sodium salt from *Pinus sylvestris* (total resin salts > 95%; three main components: Abietic acid (48.1 g/100 g), Dehydroabietic acid (21.8 g/100 g), Palustric acid (7.6 g/100 g), Forchem Oy, Rauma, Finland). Resinate stock solution was prepared from the resinate powder (40 mg RS/mL H_2_O).

### 2.1. Growth Inhibition of Planktonic Cells

To test the effects of RS on the viability of *S. mutans*, the bacteria were first grown overnight in BHI medium (Becton, Dickinson Co., Buena, NJ, USA) at 37 °C, after which they were washed twice with phosphate-buffered saline (PBS, pH 7.4) and adjusted to OD_550nm_ = 0.35 in PBS. Sodium resinate was serially diluted (0–100 µg/mL in BHI) in the rows of the 96-well plate, and bacterial suspension (1:10) was added to each well. The plates were incubated in a CO_2_ incubator (IncuSafe, Sanyo Electric Biomedical Co., Osaka, Japan) at 37 °C, and the OD_594nm_ was measured at indicated intervals. For the measurement, the plate was moved from the incubator to the microplate reader (Multiscan FC, Thermo Scientific, Shanghai, China) in normoxia. To avoid bacterial precipitation, samples were thoroughly mixed before OD measurements. Immediately after reading, the plate was returned to the incubator. Growth inhibition was measured in two independent assays, both containing eight replicates of each concentration. Results are expressed as means of the replicates.

### 2.2. Viability

The bacteria were prepared as described in Section 2.1, mixed with pre-warmed sodium resinate dilutions, and incubated at 37 °C. After 2, 15, and 60 min of incubation, an aliquot of 100 µL was withdrawn and used for CFU counting on agar plates. Plates were incubated for 2 days in a CO_2_ incubator at 37 °C before colonies were counted. Experiments were run in triplicates and repeated in two independent assays.

### 2.3. Effects of RS on Biofilm Formation

Because 2 min exposure to resinate did not kill the bacteria, we tested the biofilm formation by the pre-treated *S. mutans* cells. Bacteria were grown overnight at 37 °C, washed with saline, and adjusted to OD_550nm_ = 0.5. Bacteria from 1 mL of this suspension were pelleted (10,000× *g*, 10 min) and re-suspended into 0.5 mL RS dilutions (0–200 µg/mL). After 2 min incubation at 37 °C, 9.5 mL saline was added to dilute the RS concentration to 1:10 of the original before the biofilm assays in 96-well plates. To induce extracellular polysaccharide formation and efficient biofilm accumulation, Jordan’s broth (containing per liter 5 g Trypticase, 5 g Yeast extract, 5 g K2HPO4, 2 g glucose, and 0.5 mL salt solution (0.8 g MgSO_4_ 7H_2_O, 0.04 g FeSO_4_ 7H_2_O, 0.019 g MnCl_2_ in 100 mL of distilled water) supplemented with 0.3% sucrose was used as a growth medium in all biofilm assays. Each well contained 180 µL of growth medium, and pre-treated bacteria (20 µL) were added. The final RS concentration presence during the biofilm formation was thus 1:100 of the pre-treatment concentration. Control wells with equal concentrations (i.e., 0–2 µg/mL) of resinate in growth media were inoculated without pre-treating the bacteria. After 8 h incubation at 37 °C in CO_2_-enriched atmosphere, the medium was removed, and the wells were carefully washed twice with PBS. Biofilms were fixed for 15 min with methanol and stained with crystal violet [32] (with some modifications). Briefly, 0.1% crystal violet was added to the wells and after 10 min the stain was removed, and the wells were washed with PBS. The remaining stain was dissolved into 33% Acetic acid and the absorbance was measured at A_594nm_. The biofilm formation was calculated as % from 0 µg/mL control ((A569 test/A569 contr) × 100).

To test the effects of RS pre-treatment on bacterial acid production during the biofilm formation, biofilms were allowed to accumulate on 24-well Hydrodish plates with integrated optical pH sensors (PreSens Inc., Regensburg, Germany) instead of 96-well plates. The total volume of the samples was 2 mL (200 µL of pre-treated bacteria, 1.8 mL growth medium supplemented with 0.3% sucrose). Sensor dishes contained pre-calibrated pH sensors integrated at the bottom of each well and were read through the transparent bottom of the dish by PreSens Sensor Dish Reader (PreSens Inc.). The fiberoptic pH sensors have a measurement range from 5.0 to 8.5. When biofilm accumulates on the sensors, the sensors record the pH of the biofilm without disrupting the biofilm structure. The plates were incubated in CO_2_-enriched atmosphere at 37 °C for 24 h and the pH was recorded every 15 min.

### 2.4. Effects of RS on Acid Production in Matured Biofilms

To test the effects of RS on *S. mutans* acid production in existing biofilms, bacteria were allowed to form biofilms on Hydrodish plates (PreSens Inc.) in the absence of RS (200 µL of bacteria, 1.8 mL Jordan’s broth supplemented with 0.3% sucrose). After 24 h, the medium was removed and the biofilms in each well were washed twice for 15 min with PBS to neutralize the pH inside the biofilm.

A serial dilution of RS was made in sucrose solution. RS dilutions were made in saline and mixed 1:1 with 2% sucrose in saline. Final RS concentrations were 0 µg/mL, 25 µg/mL, 100 µg/mL, and 200 µg/mL in 1% sucrose. After pH neutralization, 1 mL of sucrose–RS was added into each well and the pH was measured at 1 min intervals for the next 45 min.

### 2.5. Statistical Analysis

The means and standard deviation for both strains and each RS concentration were obtained from the data. Differences between strains and RS concentrations were tested for significance by two-way ANOVA or by Student’s *t*-test.

## 3. Results

### 3.1. Growth Inhibition of Planktonic Cells

Sodium resinate inhibited the growth of both *S. mutans* strains in a dose-dependent manner (Figure 1) The type strain was inhibited by concentrations ≥ 6.25 µg/mL and was slightly more sensitive than the clinical isolate that was inhibited by concentrations ≥ 12.5 µg/mL. The concentration of 100 µg/mL prevented the growth of both strains completely in the course of the 8 h follow-up.

### 3.2. Viability

The treatment of the bacteria with high concentrations of RS killed the cells/reduced their growth in a time- and dose-dependent manner (Figure 2). Incubation with ≥100 µg/mL for 60 min appeared to kill all bacteria, while the short 2 min exposure to 100 µg/mL slightly reduced the number of growing *S. mutans* bacteria. The clinical isolate CI2366 was more sensitive to short-time exposure. For example, after 2 min treatment with 200 µg/mL, no growth was detected from the clinical isolate CI2366 while a logCFU value of 4.9 (±0.59) was recorded for the type strain NCTC10449.

### 3.3. Effects of RS on Biofilm Formation

Because short pre-treatment appeared to affect the bacterial viability only in high concentrations, we tested the effects of the 2 min pre-treatment on the ability of bacteria to form biofilm after RS was removed/diluted away. Under the conditions used, the clinical isolate produced less biofilm, but both strains produced measurable biofilm in 8 h (A569 2.08 ± 0.07 for NCTC10449, 0.523 ± 0.08 for CI2366). As expected, the pre-treatment with the high concentrations that affected the viability also reduced the biofilm formation, but some reduction in the biofilm formation was seen with all tested concentrations (25–200 µg/mL, Figure 3). Strain CI2366 appeared to be more sensitive than the type strain. Because the RS was only diluted, not removed, after pre-treatment, control experiments with the presence of diluted RS were performed. The presence of the highest concentration (2 µg/mL; 1:100 dil of 200 µg/mL) of RS reduced the biofilm formation by the CI2366 strain (A_569_ 0.470 ± 0.04 vs. 0.401 ± 0.03, *p* < 0.05) even without pre-treatment, while a lower concentration did not affect the biofilm formation without pre-treatment (e.g., 1 µg/mL A_569_ 0.455 ± 0.03, *p* = 0.15).

We also followed the acid production during the biofilm formation on Hydrodish plates where the pH can be monitored in real time. The pH levels remained near neutral for 5 to 7 h, after which they started dropping rapidly. The lowest pH that the optical sensor can measure is pH 5, and in every well with strain NCTC10449 pH 5 was reached after 8 h and 45 min, with a 60 min difference between RS concentrations 0 and 200 µg/mL (Figure 4A). This pH was significantly different between 0 and 200 µg/mL from 5 h 15 min (*p* < 0.05), and between 0 and 100 µg/mL from 6 h 45 min (*p* < 0.05, paired *t*-test). For strain CI2366, it took appr 12.5 hours to reach a pH level of 5, and the difference between 0 and 200 µg/mL was significant (*p* < 0.05 from time-point of 7 h, Figure 4B).

### 3.4. Effects of RS on Acid Production in Matured Biofilms

The *S. mutans* biofilm was allowed to accumulate in Jordan’s broth supplemented with 0.3% sucrose for 24 h, after which the medium was removed and the pH inside the biofilms was neutralized with PBS. A solution of 1% sucrose with different concentrations of RS was added on the biofilms and the decrease in pH was followed. After the application of sucrose, the pH immediately started to decrease, but the presence of RS significantly retarded the pH decrease for both strains (Figure 5, *p* < 0.05, ANOVA), and the effect was dose-dependent.

## 4. Discussion

The aim of this study was to determine the effects of pine resinate (RS) on the viability, growth, and biofilm formation as well as acid production of cariogenic *S. mutans* strains in developing and mature biofilm. The hypothesis was that RS reduces growth and inhibits acid production of *S. mutans* in planktonic form but also in biofilms.

Earlier studies of the rosin effects on oral bacteria focused mainly on bacterial growth [26,28,33]. They used resin extracts from *Vigueira arenari* [28,33] or pure abietic acids [26]. Carvalho et al. suggested that salts from acidic diterpenes could increase antimicrobial capabilities compared with acids [33]. In our study, we used resin salts extracted from *Pinus sylvestris* and found them effective in reducing bacterial growth in a time- and dose-dependent manner. Compared to earlier studies of the abietic acid effects on *S. mutans*, similar concentrations of RS prevented the growth of bacteria as reported for pure abietic acid (25–50 µg/mL vs. 64 µg/mL, respectively). On the other hand, Porto et al. described the antimicrobial properties of various purified and semisynthetic diterpenes from *V. arenaria* with efficient antimicrobial potential against *S. mutans* with MIC values below 10 μg/mL [28]. Abietic acid is a major component in pine-oil-derived resin. RS, used in the current study, is a mixture of molecules, and, according to the manufacturer, contains 48.1% salts of abietic acid. It might be that the salts are slightly more effective than the abietic acid, but additional components in the RS preparation may have high antimicrobial potential as well. In addition, we detected differences between the *S. mutans* strains, which might also contribute to the differences reported in different studies. Thus, resinates have significant antimicrobial potential, but further studies regarding the differences between resin acids and resinates are required.

The antimicrobial effects of the RS were treatment-time-dependent. Tooth brushing or rinsing with mouthwashes is recommended to last 2 min, and in our experiments, the high resinate concentrations completely prevented bacterial growth after 2 min incubation. Short, 1–5 min exposure times of bacteria causing hospital-acquired infections (*Enterococcus fecium, Staphylococcus aureus, Klebsiella pneumoniae, Acinetobacter baumanii,* and *Pseudomonas aeruginosa*) to rosin are shown to be enough to effectively induce the killing of these bacteria [34]. Gram -ve *A. baumannii* and gram +ve *E. fecium* were equally sensitive to being killed by 5% rosin after 1 min exposure. Gram -ve *K. pneumonaie* was the most resistant, and only a slight reduction in growth was seen after 5 min exposure [34]. Such short exposure times have not been studied before with *S. mutans.* For example, in their study, Ito et al. showed reduced *S. mutans* biofilm formation in the presence of pure abietic acid and a reduced number of colony-forming units recovered from biofilm after 1 h of exposure to abietic acid [26]. In the oral cavity, the external substances rarely remain for longer periods because they are rapidly flushed away with saliva and removed by swallowing. Therefore, a substance to be effective in the oral cavity should either have a high affinity to dental/oral surfaces or act in a short time, as noted for RS in the current study. Interestingly, Bell et al. noted that by introducing an organic contaminant in the form of BSA, the antimicrobial effects of rosin were slightly changed. *E. faecium* was again killed after 1 min exposure but most of the other bacteria exhibited reduced susceptibility [34]. Only *K. pneumoniae* showed enhanced susceptibility in these conditions. Such organic contaminant can be speculated to mimic the situation in saliva, where saliva proteins provide the organic supplement in the environment. The effect of saliva on the antimicrobial activity of RS remains to be studied.

In line with the noted reduced viability, short exposure to higher concentrations (100–200 µg/mL) of RS had a pronounced inhibitory effect on biofilm formation and acid production. Short exposures to lower concentrations of the resinate did not reduce the viability but retarded the biofilm formation and, importantly, reduced the acid production of the treated bacterial population. Ito et al. showed that the presence of abietic acid in 16 µg/mL concentration can reduce *S. mutans* acid production but does not affect the biofilm formation [26]. In our assay, however, the resinate was removed/diluted after 2 min exposure, and yet the acid production and biofilm formation were reduced for several hours. Further studies are needed to reveal the antimicrobial mechanisms of resinate, but resin acids are known to disrupt the bacterial cell membrane, leading to defects in transcription and gene expression [35,36]. Similarly, rosin derivatives are suggested to disrupt the phospholipid bilayer of the viral envelope [34]. Even though the mechanism of the disruption is not known, the resin derivatives are suggested to act as surfactants. Morphological changes of the *S. mutans* cell wall are seen after abietic acid treatment even with low concentrations that did not affect the viability [26]. Such non-lethal membrane defects may also explain the reduced acid production after resinate treatment.

On Hydrodish plates, the optical pH sensors are located at the bottom of the wells and they allow for real-time in situ measurements of pH changes without disrupting the biofilm. Thus, they provide an interesting tool for biofilm studies. When RS was added to bacteria already existing in biofilms, the addition reduced the acid production inside the biofilms, indicating that the RS can penetrate the bacterial biofilm matrix. The RS did not completely block the acid production of *S. mutans*, and eventually in all the biofilms the pH dropped below 5.5 which is considered a critical pH for enamel demineralization. This was true even in the biofilms exposed to high RS concentrations that had strong antimicrobial effects on planktonic bacteria. This may be explained by the fact that, in addition to the protection offered by the matrix, bacteria in biofilms adapt to environmental conditions, such as nutrient limitations by reduced metabolism and differential gene expression, which may make them more resistant to antimicrobial agents [37]. Still, the recorded pH difference between the highest and zero RS concentration after 20 min was almost 1 pH unit. This can be considered as a marked difference, which might also have physiological importance. In the oral cavity, saliva is continuously flushing the biofilms. Even though the matrix reduces the penetration of salivary components into the biofilms, buffering molecules such as phosphates and especially bicarbonates can diffuse through the matrix and neutralize the pH inside the biofilms. The obtained reduced acid production can be speculated to increase the potential of acid neutralization by saliva.

RS reduced acid production inside mature biofilms of both strains of *S. mutans*, but the type strain NCTC10449 was more affected. This was an interesting finding because in all other tests the clinical isolate CI2366 appeared to be more sensitive to RS. The composition of the biofilm matrix produced by the strains is different [38], which might affect the RS penetration into the biofilm and thereby its effects on acid production. This, again, highlights the important influence that the biofilm matrix can have on the potency of antimicrobial compounds. The penetration and effects of RS in more complex multispecies biofilms in vitro or in vivo remain to be studied.

Resin acids, as well as their sodium salts, already have many industrial applications as lubricants, sealers, finger paints, etc. In the dental field, for example, abietic acid is included in the temporary sealer Plast Seal Quick (Nippon Shika Yakuhin, Japan). Many terpenes, such as abietic acid, are irritants and may cause allergic skin reactions according to REACH profile (https://echa.europa.eu/brief-profile/-/briefprofile/100.057.386 (accessed on 2 February 2024)). Resin acids and rosin acids, and their sodium salts, may also cause eye irritation, and at high concentrations (>20%) may induce skin sensitization [39]. The concentrations found to be effective in this study were much lower. Yet, before any clinical use, the effect of RS from *Pinus sylvestris* on human cells and mucosa needs to be carefully evaluated.

## 5. Conclusions

In conclusion, RS extract reduced the growth and acid production of cariogenic *S. mutans* both in the planktonic phase and inside a biofilm. In comparison to the resin acids with well-established antimicrobial activity, the more water-soluble resinates could be more suitable in dental applications. Their safety for human cells remains to be studied.

## Figures and Tables

**Figure 1 dentistry-12-00040-f001:**
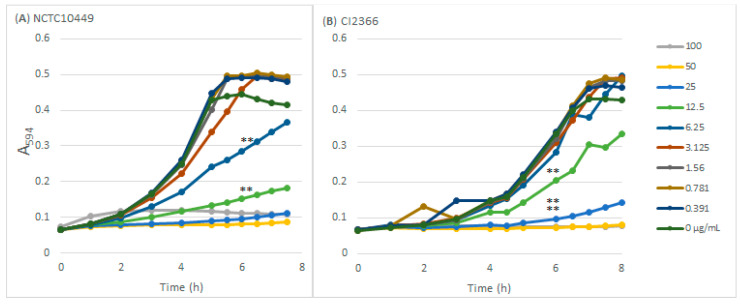
The effects of RS on *S. mutans* growth. *S. mutans* strains NCTC10449 and CI2366 were inoculated to BHI medium supplemented with different concentrations (µg/mL) of RS and the growth was followed by measuring the increase in turbidity (A594). ** *p* < 0.01 difference compared with control at 6 h time-point.

**Figure 2 dentistry-12-00040-f002:**
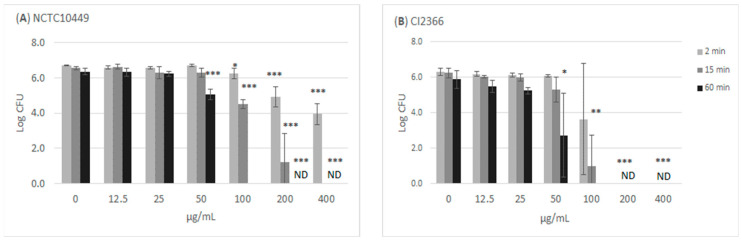
Antimicrobial effect of RS on *S. mutans*. The bacteria were mixed with different concentrations of RS. Aliquots were withdrawn at indicated time intervals, diluted in NaCl, and plated on agar plates to measure CFU. * *p* < 0.05, ** *p* < 0.01, *** *p* < 0.001 difference compared with non-treated control (paired *t*-test). ND: no colonies detected; detection limit Log (CFU/mL) = 2.

**Figure 3 dentistry-12-00040-f003:**
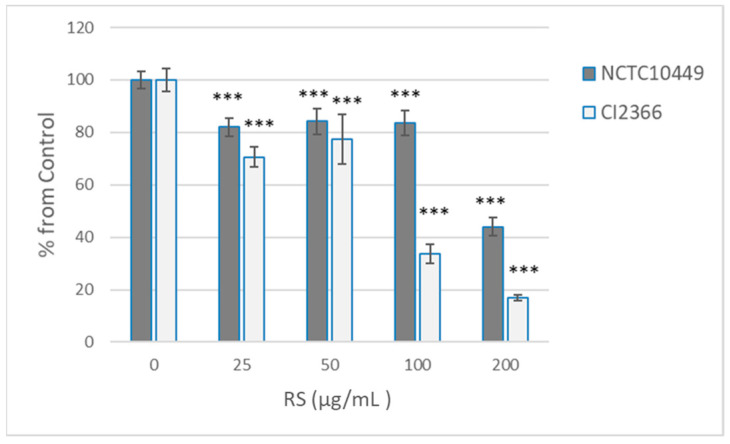
Effect of RS treatment on *S. mutans* biofilm formation. *S. mutans* strains NCTC10449 and CI2366 were incubated with indicated concentrations of RS. After 2 min, the suspension was added 1:100 to the growth medium for biofilm formation. After 8 h, the biofilms were stained with crystal violet and absorbance at A569 was measured. The biofilm formation was calculated as % from 0 µg/mL control ((A569 test/A569 contr) × 100). *** *p* < 0.001 difference compared with control (0 µg/mL).

**Figure 4 dentistry-12-00040-f004:**
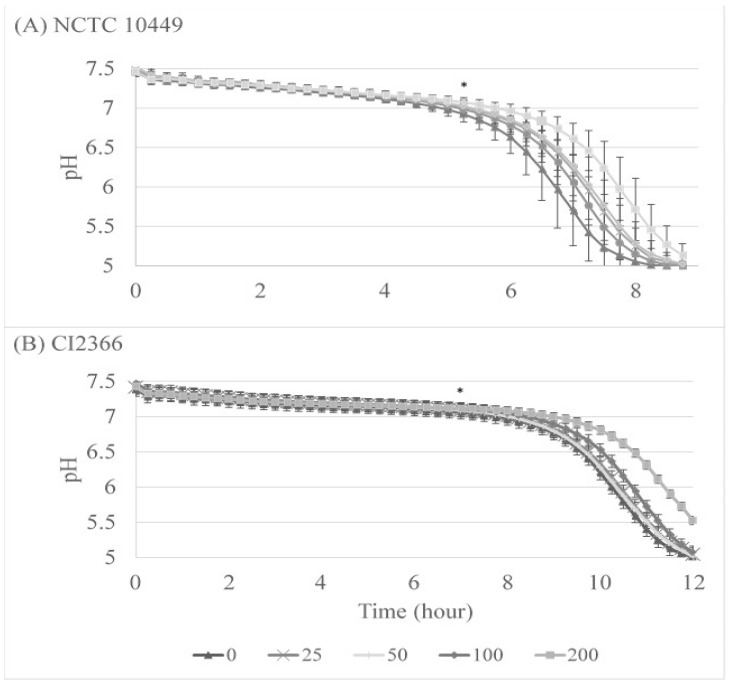
The effect of RS on S. mutans acid production during biofilm formation. *S. mutans* strains NCTC10449 and CI2366 were incubated with indicated concentrations (μg/mL) of RS. After 2 min, the suspension was added 1:100 to the growth medium for biofilm formation. pH was measured every 15 min. * First time-point for significant difference between test and 0 µg/mL (*p* < 0.5, paired *t*-test).

**Figure 5 dentistry-12-00040-f005:**
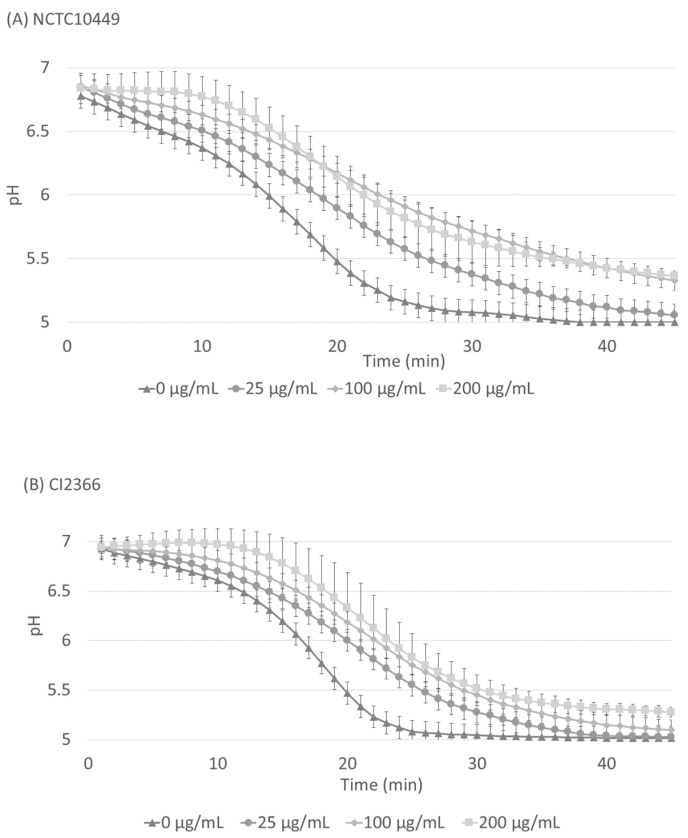
Acid production in matured biofilms after sucrose exposure. *S. mutans* strains NCTC10449 and CI2366 were allowed to form biofilms in Hydrodish wells. After 24 h, the biofilms were washed and neutralized with PBS and exposed to 1% sucrose supplemented with different concentrations of RS. The pH was measured every minute for 45 min. In the results of both strains there was a statistically significant difference (*p* < 0.008) between RS 0 µg/mL and 200 µg/mL (paired *t*-test, Bonferroni correction).

## Data Availability

All created data are available from the corresponding author upon reasonable request.

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
