# Peer review of "Pine-Oil-Derived Sodium Resinate Inhibits Growth and Acid Production of Streptococcus mutans In Vitro"

_dentistry, 2024, doi:10.3390/dj12020040_

Round 1

Reviewer 1 Report

Comments and Suggestions for Authors

Rajala et al have studied the effects of Pine Oil Derived Sodium Resinate on the cariogenic bacterium Streptococcus mutans. The authors have found that the resinate was able to reduce bacteria growth and biofilm formation in addition to reducing the acidity in the biofilm. These are my comments to the authors.

Is there an IRB approval for the clinical sample taken?

2. Materials and methods, S. mutans was taken directly from the frozen stock to liquid broth which is not the proper way and is a faulty method. It is supposed to be steaked on agar first and then a single isolated colony matching the colony morphology of S. mutans must then be transfered to broth to insure no contamination in the media. 

Do we know if it is safe for Pine oil derived sodium resinate to be used on humans at the concentrations used in this study? if so, please add that to the introduction. If no, then an experiment demonstrating the safety on human cell culture needs to be added. 

What type of media was S. mutans gown in after thawing? was it put directly into BHI? How was the bacteria isolated from the patients? what steps were taken to isolate S. mutans from all the other strains in the oral cavity? What was done to insure that it was S. mutans

2.1 What devices were used to insure bacteria were grown in CO2 enriched environment? Was there a transfer step between the incubator and the plate reader? Or was it grown in the plate reader? How was the CO2 supplied if it was incubated in the plate reader? 

For this statement Sodium resinate was serially diluted (0-100 µg/ml in growth medium)" I assume the growth medium is BHI? if it is please change the word growth medium to the actual medium name. 

2.2 again name of the medium was not mentioned. Please mention in it. Please mention the type of incubator used

2.3 Again medium name not mentioned. Please mention it. If the medium is BHI, BHI already has glucose in it, what is the purpose of adding 0.3% sucrose? Why was the bacteria adjusted to OD550nm to 0.5 here while it was 0.35 in the growth inhibition study? 

I do not understand the purpose of testing biofilm growth 8 hours later. 8 hours is usually not even enough to get a good turbidity growth in broth, nonetheless biofilm formation. S. mutans is not a fast grower, you need much more time to get a decent biofilm read. 

Effects of Pre treatment on biofilm formation, it is hard to see the picture without knowing what type of media was used. Either way, what is the rational behind this experiment? When comparing it to the 2.4 experiment it seems redundant. Could you please explain why you need both? It is understandable that acidity plays an important role in dental caries, but these 2 experiments are delt with in a strange way. Having on as a part of 2.3 in with other experiments, then testing acidity again in a separate section 2.4. 

2.4 why was 1% sucrose used here vs 0.3% in previous steps?

Biofilm experiment should be repeated to include 24 and\or 48 hour biofilms

Please add references for all the protocols used.

Please correct some bacterial names which were not italicized. 

Please add titles to all the graphs 

Figure 1: Growth curves are done in log format. Please correct the image to be presented as a growth curve and not just turbidity which is the most agreed upon method to test effects on bacterial growth. 

Figure 2 please mention the name of the medium used. 

Figure 3: please add the name of the medium used

Figure 4: Was the reduction in acidity statistically significant? The reduction in acidity is expected due to the fact your previous experiment showed that there was a reduction in biofilm. Less bacteria = less biofilm = less acidity. Did you factor in the fact there the biofilm was reduced when you did the measurement? 

Final thoughts to the authors. The study is weak in terms of microbiological work as the protocols for bacterial handling during thawing and initial growth is faulty. The conditions of the experiments vary from one experiment to the next which makes the whole study asymmetric. The lack of proper microbiological guidance shows in the prep step and also in the growth inhibition study where it was presented as a growth curve. The information in terms of the effects of the resinate is not bad in it self, but the means by which the information was gathered is the problem.  

Comments on the Quality of English Language

All bacterial names need to be italicized. Other than that the language is pretty sound .

Reviewer 2 Report

Comments and Suggestions for Authors

The assessed work is very interesting. The tests are performed correctly and the results are medically significant. Unfortunately, before publishing, I propose corrections:

1. A phytochemical test (e.g. HPLC, TLC) of pine oil resinate should be performed and the percentage composition should be provided.

2. The Discussion should describe the cytotoxicity of the compounds obtained from the phytochemical test. This is important because many terpenes have cytotoxic effects. If the literature confirms the toxic effects of the ingredients of pine oil resinate, then the conclusions in the paper will change.

3. Many references are old. You should add publications from the last 2-3 years.

Reviewer 3 Report

Comments and Suggestions for Authors

Thank you for selecting me as a reviewer for the article entitled “Pine Oil Derived Sodium Resinate Inhibits Growth and Acid Production of Streptococcus mutans in vitro” submitted to Dentistry Journal. This study was conducted to determine the effect of pine oil derived sodium resinate on the growth, biofilm formation, and production of acid from the dental pathogen, Streptococcus mutans. The authors aim to expand on a subset of literature that investigates these properties of substances found in nature with the potential for therapeutic or prophylactic benefit for oral disease. The gap in knowledge they aimed to address was to use a water-soluble form of resin acid salt which would be more convenient than the acids used in previous studies.

The authors systematically addressed the dosing and time dependency of the product on planktonic growth of a type strain of Streptococcus mutans, as well as a clinical isolate from a previous study. Additionally, they showed the bactericidal activity at shorter time points to reveal that the compound is able to kill Streptococcus mutans at higher concentrations rather than just inhibiting growth.

Furthermore, they tested pretreatment of S. mutans with various concentrations of the sodium resinate to mimic the time period of mouthrinsing, and then evaluated the subsequent ability of biofilm formation and acid production during formation and when RS is added to a mature biofilm. While there was a substantial decrease in biofilm size after 200 ug/ml concentration pretreatment, there were only modest, yet statistically significant decreases in biofilm size with pretreatment at concentrations lower than 100 ug/ml in the type strain. The effect was much larger in the clinical isolate, which by the y-axis in figure 3 produces substantially less biofilm than the type strain.

Addition of RS with sucrose to mature biofilms was a good addition to the study, however, the relevance of the change in time to pH drop is not addressed. While mathematically significant, all of the treatments still reach a pH level below the threshold of enamel demineralization, albeit, about 10-15 minutes later. Is this a physiologically relevant difference? Some discussion on this difference could be added to the discussion, or it could use a compound that is known to prevent pH drop as a control.

Overall, I think this article is suitable for publication and was a thorough investigation of a water-soluble sodium resinate for comparison to previous articles that utilized other acid forms with less solubility.

Major concerns:

I do not have any major concerns over the article or methodology which was all very logical and appropriate. However, I do have small concerns that should be addressed in terms of figure quality.

Minor Comments

Overall:

“Streptococcus mutans” needs to be italicized throughout the article and there were some issues with numbers not being properly superscript/subscript.

All figures have low resolution issues and need to be addressed

Materials and Methods:

Lines 76-77: add the equipment and manufacturer for the plate reader

Results:

Figure 1: While not necessary since the information is in the legend, putting the strain name on top of the graphs would make them much easier to evaluate

Figure 2: The use of the * or multiple *** makes it difficult to read the graph and understand the comparisons. I would recommend using a single star to denote significance. Also, statistical tests used should be listed in the legend. IF there were no colonies recovered, then the bar should read below detection and the minimum level counted should be listed in the figure legend.

Figure 2a: the 100 ug/ml treatment of the type strain is marked as statistically significant. Is this a mistake or an actual result. If true, is this a physiologically relevant difference or potentially an arbitrary difference due to the study design where there are 8 replicates per independent experiment.

Line 160: control concentrations without pretreatment should be shown as well in the figures

Figure 3: Figure is not high enough resolution and is blurry. I would recommend axis titles on both panels even if they are the same

Figure 3: The axes on both panels are not equal and it is clear that both strains produce biofilms at significantly different levels. I think that both strains should be shown on the same graph or at least that the axes should be the same range. Also, consider including % reduction of biofilm so that each concentration can more easily be compared by readers.

Figure 3: The comparisons with the diluted RS that was included in the biofilm growth media should be added to the presented graphs.

Line 172: “even more evident”, was there a statistical analysis done on these data to show differences?

Figure 4: x-axis is in min but in the text is being reported mostly in hours. Were any stats performed on this experiment. Even if not significant, that should be reported

Figure 5: legend has p-values displayed with comma instead of decimal

Discussion:

The pH drop from the mature biofilm is clearly decreased with the addition of RS into the sucrose solution. However, it is not addressed whether this is a difference with the physiological state of the biofilm-associated bacteria, or if the bacteria are being killed with the exposure. I think this should be touched on in the discussion a bit more.

Crystal violet shows the overall biofilm production, but is there any insight as to the composition of the biofilm in terms of extracellular matrix and bacteria? It was mentioned in the discussion that the matrix composition is different in the clinical isolate compared to the type strain, but would be interesting to know what the difference is after RS treatment

Round 2

Reviewer 1 Report

Comments and Suggestions for Authors

First of all I appreciate the time and effort it took the authors to respond to the comments and the attempts to correct the parts that need corrections.

I have a major concern about the protocol of taking bacteria straight from the cryovial to the broth without first growing it on agar. There might have been a contamination during freezing or during inoculation. Without taking a freshly grown isolated colony there is no way to confirm the presence of the proper strain of bacteria. The fact that this has been practiced for a while does not make it better if not worse. There is the fact that bacteria need to be in the log phase as well which was not mentioned as well. You should only work with a clearly isolated colony and in mid-log phase. however, despite the technical errors it does not nullify the results that you have acquired. I recommend better care in terms of bacteria handling, especially in the initial phases after thawing. The results are good and I believe good information can be extracted from the work done here. 

Reviewer 2 Report

Comments and Suggestions for Authors

Excellent corrections. Thank you